# Chemical Profile and Screening of Bioactive Metabolites of *Rindera graeca* (A. DC.) Bois. & Heldr. (Boraginaceae) In Vitro Cultures

**DOI:** 10.3390/plants10050834

**Published:** 2021-04-21

**Authors:** Konstantia Graikou, Harilaos Damianakos, Christos Ganos, Katarzyna Sykłowska-Baranek, Małgorzata Jeziorek, Agnieszka Pietrosiuk, Christos Roussakis, Ioanna Chinou

**Affiliations:** 1Lab of Pharmacognosy and Chemistry of Natural Products, Department of Pharmacy, National & Kapodistrian University of Athens, Zografou, 15771 Athens, Greece; kgraikou@pharm.uoa.gr (K.G.); harisdam@pharm.uoa.gr (H.D.); chris50ganos@hotmail.com (C.G.); 2Department of Pharmaceutical Biology and Medicinal Plant Biotechnology, Faculty of Pharmacy, Medical University of Warsaw, 1 Banacha, 02-097 Warsaw, Poland; kasiasb@farm.amwaw.edu.pl (K.S.-B.); gosiajeziorek@op.pl (M.J.); agnieszka.pietrosiuk@wum.edu.pl (A.P.); 3IICi MED/EA 1155- Dept Cancer du Poumon et Cbles Moleculaires, UFR Sciences Pharmaceutiques- 9 rue Bias, CEDEX 1, 44035 Nantes, France; christos.roussakis@univ-nantes.fr

**Keywords:** Boraginaceae, *Rindera graeca*, in vitro culture, hairy roots, caffeic acid derivatives, quercetin 3-rutinoside-7-rhamnoside, pyrrolizidine alkaloids, rinderol, antimicrobial activities, antiproliferative activities

## Abstract

*Rindera graeca* is a rare endemic plant where in vitro culture has been used in order to investigate bioactive metabolites. Phytochemical study of the in vitro shoots and hairy roots led to the isolation of seven phenolic derivatives and the unusual furano-naphthoquinone rinderol. *R. graeca* was also analyzed for its pyrrolizidine alkaloids content by LC-MS, and it was found to contain echinatine together with echinatine and rinderine N-oxides. Rinderol, isolated only from in vitro hairy root culture for the first time in the genus, revealed promising bioactivities. It was evaluated in vitro against a panel of microorganisms, showing very strong activity specifically against Gram-positive bacteria (MIC values 0.98 × 10^−2^–1.18 µg/mL) as well as very interesting antiproliferative effect against the human non-small-cell bronchopulmonary carcinoma cell line NSCLC-N6-L16 and the epidermoid lung cancer cell line A549. These findings were compared with the chemical profile of the plant from nature, while this study is the first to report on the effects of *R. graeca* extracts obtained from in vitro culture, providing a valuable contribution to the scientific community towards this sustainable method of production of potential bioactive molecules.

## 1. Introduction

In the framework of the studies of our scientific team on Mediterranean endemic Boraginaceae species [1,2,3,4,5,6,7,8,9], the genus *Rindera* Pall. belonging to tribe Cynoglosseae of the borage plant family has attracted our scientific interest [8]. The genus *Rindera* is comprised of 25 species [10], and is widespread mainly in Europe and Asia. It is characterized as a source of caffeic acid derivatives, flavonoids, and pyrrolizidine alkaloids (PAs) while its seeds are a rich source of fatty acids.

*Rindera graeca* Boiss & Heldr. has been studied phytochemically very recently, while its antioxidative properties have been evaluated showing an interesting chemical profile and bioactive potency [8].

As *Rindera* is a rare species, in this study we were focused on the alternative modern approach of its in vitro culture and towards the phytochemical study of its crude material, which was further analyzed and presented herein. Preliminary in vitro cultures on its roots led to the detection of naphthoquinone shikonin-type metabolites, revealing it as a rich source of caffeic acid derivatives (rosmarinic, lithospermic acids, etc.) [1].

According to the literature, several biotechnological approaches have been used to enhance the production of phenolic compounds in in vitro cultures of plant organs, tissues, and cells [11], especially towards rosmarinic acid (RA) production, which is a well-known chemotaxonomic marker among Boraginaceae plants [7,8,9]. RA, as an ester of caffeic acid, is often followed by its derivatives, lithospermic acid (LA) and rabdosiin [5,8].

*Rindera graeca* and *Rindera gymnandra* are the only endemic Mediterranean taxa in Greece and Algeria, respectively [12]. Other *Rindera* species endemic in Turkey (*R. dumanii*, *R. caespitosa*, and *R. cetineriare*) have been used in folk medicine, appreciated for their anti-inflammatory properties [13]. The essential oil from *R. lanata* var. *canescens* showed moderate in vitro antimicrobial activity [14], while a methanolic extract of *R. lanata* var. *lanata* exerted in vitro antivirus property against a human rotavirus [13]. The fruits of the widely distributed *R. oblongifolia* are a rich source of unsaturated fatty acids [15], similar to the aerial parts, roots, and seeds of the endemic Serbian *R. umbellata*. A number of pyrrolizidine alkaloids (PAs) have been isolated from the same species [10].

PAs are plant toxins associated with disease in livestock and pose a serious health risk to humans, as they may enter the food chain of humans via meat, milk products, and herbal teas [8,16].They are estimated to be produced by 3% of all flowering species [2] in the form of bases and pyrrolizidine alkaloid N-oxides (PANOs) [17], while they are among the most common secondary metabolites along with phenolic metabolites and naphthazarin pigments in the Boraginaceae family [4], Cynoglossae tribe [2], particularly the *Rindera* genus [8,18,19].

The increasing market demand for bioactive extracts and pure metabolites of herbal origin led various biotechnological methods and techniques to be further developed towards the production of active secondary metabolites, protecting both the sustainability and biodiversity of the plant kingdom, which is currently of high importance [20]. The aim of this study was to analyze the plant material derived from in vitro cultures of the rare endemic plant *Rindera graeca*, evaluating its phytochemical potential for the production of phenolic compounds, to determine the PAs content, and to compare the results with those obtained in a recent study of the naturally occurring plant [8].

Furthermore, we evaluated the antimicrobial and cytotoxic properties of plant extracts obtained via biotechnological methods, together with the most promising pure metabolites.

## 2. Results and Discussion

### 2.1. Identification and Isolation of Secondary Metabolites

Chromatographic separation led to the identification of eight metabolites (Table 1): one flavonoid triglycoside (quercetin-3-rutinoside-7-rhamnoside) (Figure 1), together with caffeic acid and its derivatives: chlorogenic acid, rosmarinic acid, lithospermic acid, rabdosiin, and disodium rabdosiin salt, as well as the furano-naphthoquinone rinderol (Figure 1). All of these were structurally determined by NMR and compared with literature data and previously isolated metabolites. Rinderol was identified for the first time in *Rindera* in vitro cultures [1], while it has previously been determined through modern spectral means from in vitro roots of *Cynoglossum columnae* (Boraginaceae) [6] by our team, and from natural roots of *Onosma paniculata*, proposed as naphthofuranin B [21].

All phenolic metabolites have been identified recently from natural plant samples of *Rindera graeca* [8]. Most of the caffeic acid derivatives have been identified in multiple Boraginaceae species, while rosmarinic acid is a well-known chemotaxonomic marker of the family [7,8,9]. Rabdosiin, a dimer of rosmarinic acid, and its disodium salt, a nontrivial compound recently isolated from *Alkanna sfikasiana* [5] and detected in natural plant of *R. graeca* [8], has previously been studied for potential cytotoxic and antiviral activities [11]. All plant material obtained by in vitro culture was also analyzed for its pyrrolizidine alkaloid content by LC-MS, following the standard BfR procedure, and it was found to contain echinatine and echinatine N-oxide (Figure 2), which have been identified from many previously studied *Rindera* species [8,18,19]. Rinderine N-oxide (Figure 2) was also identified, which has previously been found in plants of the Cynoglossae tribe.

Comparing the results of the chemical analyses between cultivated in vitro roots and shoots as well as recently studied naturally occurring aerial parts of *R. graeca* [8] (Table 1), it was revealed that the pyrrolizidine alkaloid content was identical in all three, showing that the in vitro cultures are not Pas free for further application.

Rosmarinic acid, a metabolite with rich biological activity, is commonly found in plants of the Boraginaceae family, serving as a chemotaxonomic marker [7,8,9]. The metabolites quercetin-3-rutinoside-7-rhamnoside and rabdosiin disodium salt, previously identified from the natural plant, were also isolated.

### 2.2. Cytotoxic Effects

The cytotoxic activity of rinderol (Figure 1), as well as n-hexane extracts derived from post-culture media of *R. graeca* root cultures, was determined against two human lung cancer cell lines: the human non-small-cell bronchopulmonary carcinoma NSCLC-N6-L16 and the epidermoid lung cancer A549. We chose these lines because human non-small-cell bronchopulmonary cancer is the most common type of lung cancer, and because of its difficulty of treatment. Non-small-cell lung cancer has a much slower proliferation time than small-cell lung cancer, which makes chemotherapy very ineffective. In addition, is a type that is difficult to detect and readily gives metastasis, which makes surgery almost impossible. The IC50 (50% inhibitory concentration test) showed that rinderol had a very clear growth inhibition capacity, with an IC50 of 1.2 μg/mL against the NSCLC-N6-L16 line and 0.9 μg/mL against the A549 line, respectively (Table 2).

Therefore, in comparison with the tested extracts, rinderol showed high cytotoxicity towards both examined cell lines (Table 2), while the IC50 value of vinorelbine (control) amounted to 0.04 and 0.001 µg/mL in A549 and NSCLC-N6-L16 cells, respectively.

The n-hexane extract of post-culture medium displayed cytotoxic potential (IC50 values 12.7 and 13.6 µg/mL, respectively). The cytotoxic activity of the methanol extract of in vitro shoots showed weaker activity (IC50 16.6 and 22.7 µg/mL, respectively). Rinderol exhibited comparable cytotoxic activity to previous results [6] against HCT-116 (colorectal carcinoma), HL-60 (human promyelocytic leukemia), and HeLa (cervical cancer), as well as comparable cytotoxicity to acetylshikonin, which revealed IC50 values against human hepatocellular carcinoma and mouse Lewis lung carcinoma cell lines ranging from 2.72 ± 0.38 to 6.82 ± 1.5 µg/mL [22]. Moreover, it was recently reported that shikonin in human A549 lung cancer cells (1–2.5 µg/mL) reduced cell viability, while at concentrations of 5–10 µg/mL it induced apoptosis [23], confirming the antitumor potential of naphthoquinone chemical structures.

### 2.3. Continuous Growth Kinetics

Furthermore, continuous growth kinetics was used to determine the mode of action of rinderol against NSCLC-N6-L16 and A549 lines treated with pure metabolite. The results of continuous kinetics on the NSCLC-N6-L16 line (Figure 3a) showed a slowdown in cell growth in a dose-dependent manner and then a complete cessation of growth during treatment with rinderol. These effects were less visible on the A549 line, but there was a significant effect on cell growth between treatment at 0.75 µg/mL and that at 1 µg/mL. The continuous growth kinetics for the NSCLC-N6-L16 line showed a profile of cytostatic antiproliferative activity (Figure 3a), while the treatment with rinderol on the A549 line showed a non-cytostatic antiproliferative activity (Figure 3b). This clearly proves that the tested molecule cannot act on the same signaling pathways for the two lines.

### 2.4. Antimicrobial Activity

The antibacterial activity of all extracts and the isolated furane-naphthoquinone rinderol was examined against a panel of six Gram (−) and Gram (+) human pathogenic bacteria strains, as well as three fungi of the *Candida* genus. The strongest activity was revealed from the pure red pigment of rinderol. This was followed by the n-hexane extracts from cultivated in vitro roots (post-culture media), which was a rich source of rinderol, and the third-highest activity was exhibited by the MeOH extracts (from in vitro shoots and natural aerial parts). The highest activity was observed against Gram-positive strains (*Staphylococcus aureus* and *Staphylococcus epidermidis* (MIC 0.98 × 10^−2^–1.18 µg/mL) (Table 3).

## 3. Materials and Methods

### 3.1. Standards and Chemicals

Analytical reagents were commercially obtained from Sigma-Aldrich (Steinheim, Germany), Fluka (Göteborg, Sweden), Acros Organics (Fair Lawn, NJ, USA), Merck (Darmstadt, Germany), and Riedel-de-Haen (Darmstadt, Germany). All chemical solvents were of HPLC grade or distilled, commercially obtained from LAB-SCAN (Dublin, Ireland), Panreac (Barcelona, Spain), Fisher-Scientific (Loughborough, UK). All chemicals and reagents, if not stated otherwise, were purchased from Sigma-Aldrich (Athens, Greece and Poznan, Poland).

### 3.2. Plant Material and Methanolic Extract Preparation

*Rindera graeca* shoot and root in vitro cultures were initiated from seeds donated by Prof I. Chinou, collected in 05/2014 from Mt. Parnon (Arcadia, Peloponnese, Greece) [8].

### 3.3. Shoot Cultures

Seedlings germinated on hormone-free, half-strength basal MS [21] solid medium (1/2 MS) in dark at 25 ± 2 °C. Next, the multiplication of shoots was carried out on solid DCR medium [24] supplemented with 0.5 mg/L 6-benzylaminopurine, in 12/12 h day/night light regime at 25 ± 2 °C. Shoot cultures were performed in 300 mL Erlenmeyer flasks containing 50 mL of the relevant medium, and plant material was transferred to fresh media every four weeks. Collected for further phytochemical investigation, shoots (Figure 4a) were lyophilized, powdered, and subjected to extraction.

### 3.4. Hairy Root Cultures

The four-week-old shoots were infected with *Agrobacterium rhizogenes* strain ATCC 15834 according to the procedure described by Pietrosiuk et al. [25]. Emerging from the places of inoculation, hairy roots were cut off and placed separately into 100 mL Erlenmeyer flasks containing 30 mL of liquid hormone-free DCR medium with the addition of cefotaxime (500 mg/L) and cultivated for two weeks to remove bacteria form their tissues. From the established 60 hairy root lines, two root lines, RgKT 7 and RgKT 17, were used for further investigations. PCR analysis confirmed the incorporation of *rolB* and *rolC* genes of *A. rhizogenes* into the root tissue. Next, hairy roots were cultivated in liquid hormone free DCR medium on horizontal shaker at 105 rpm (INFORS, Basel, Switzerland) at 23 ± 1 °C in the dark, in 250 mL Erlenmeyer flasks containing 50 mL of the medium, with subculture to fresh medium every four weeks [26,27]. Collected for further phytochemical investigation, hairy roots (Figure 4b) were lyophilized, powdered, and subjected to extraction.

### 3.5. Extract Preparation

The lyophilized and powdered plant material (shoots and hairy roots) was subjected to phytochemical investigations. Shoot extraction performed with methanol was carried out for 15 min using ultrasonic bath (Sonorex, Bandelin, Germany) and incubated overnight on an INFORS horizontal shaker (105 rpm) at 23 ± 1 °C in the dark. Afterwards, the samples were centrifugated, re-extracted using methanol, and supernatants were combined and evaporated to dryness. The root tissues were sonicated with n-hexane for 15 min multiple times to perform exhaustive extraction. The resulting root extracts were combined and evaporated to dryness. The thus-prepared shoot and root extracts were stored at −20 °C before further analysis.

### 3.6. Phytochemical Analysis

#### 3.6.1. Chromatographic Fractionation of In Vitro Shoots

An amount of 2 g of the methanolic extract was subjected to molecular weight chromatography using a 30 cm column (∅: 2 cm) with a Sephadex LH-20 (Pharmacia) stable phase and an isocratic 9:1 MeOH/CH_2_Cl_2_ mobile phase; 58 fractions were obtained. After visualization using thin layer chromatography (stable phase: cellulose plate, mobile phase: 85/15 Η_2_Ο/CH_3_COOH, visualization agent: Naturstoff), seven compounds (caffeic acid (8.4mg), chlorogenic acid (8 mg), rosmarininc acid (11 mg), lithospermic acid (5 mg), rabdosiin (3 mg), disodium rabdosiin salt (2 mg), and quercetin 3-rutinoside-7-rhamnoside (7 mg)) were detected and compared with our compound library and further purified through preparative TLC (Appendix A). Their structures were confirmed through NMR analysis and comparison with literature data [8].

#### 3.6.2. Chromatographic Fractionation of In Vitro Hairy Roots

An amount of 1.5 g of the *Rindera graeca* n-hexane extract was subjected to column chromatography on 2.5 g silica gel (0.015–0.040 mm) eluted with cyclohexane/CH_2_Cl_2_ and CH_2_Cl_2_/MeOH step gradient, yielding fourteen fractions. The fractions were chromatographed on TLC aluminum plates and the fraction rg9 (4 mg) yielded rinderol which was structurally confirmed by NMR spectroscopy (Appendix A) and comparison with literature data [6].

The purity for each isolated compound was approx. ≥90% according to NMR spectra.

#### 3.6.3. Analysis of Pyrrolizidine Alkaloids (PAs)

For the isolation and identification of PAs from *Rindera graeca* aerial parts (Appendix A), the BfR [28] method was used: 2 g of dried plant material was macerated in 20 mL H_2_SO_4_ 0.05 M for 15 min under sonication. Then, the macerate was centrifuged for 10 min at 3800 rpm and the upper liquid layer was extracted. This process was repeated once. The final extract was neutralized to pH 7 and filtrated.

The presence of PAs was confirmed through the Mattocks/Molyneux method, first applying a 10% acetic anhydride solution in benzene/naphtha 4:5 and then Ehrlich chemical agent on a silica TLC plaque which was developed in an 88:5:2 CH_2_Cl_2_/MeOH/NH_4_OH system. The PAs were visualized as purple spots [8].

#### 3.6.4. Liquid Chromatography/Mass Spectrometry (LC/MS) Analysis of PAs

The PA/PANO extract was examined by qualitative LC/MS analysis. The scientific equipment used was an Agilent 6500 Series Accurate-Mass Quadrupole Time-of-Flight (Agilent Technologies Inc., Santa Clara, CA, USA) device equipped with ESI-Jet Stream ion source and Atlantis HILIC silica column (150 × 2.1 mm, dp = 3 μm) (Waters, Milford, MS, USA). The chromatograph used a diode array detector autosampler, dual grading pump, and column heater. An RP18 stable phase was used in conjunction with a gradient elution mobile phase of 0.1% formic acid in methanol with a stable flow of 0.25 mL/min. For this particular experiment the MS scans were acquired in positive ionization mode.

#### 3.6.5. Nuclear Magnetic Resonance (NMR) Analysis

Samples of the abovementioned metabolites were dissolved in NMR-grade D_2_O and CDCl_3_ as a solvent and TMS as an internal standard utilized for ^1^H-NMR analysis, using a Bruker DRX 400 (400 MHz) instrument (Bruker, Rheinstetten, Germany). After examination of the received spectra with analysis software and comparison with available reference data, the compounds were fully determined.

### 3.7. Biological Activities

#### 3.7.1. Cytotoxicity Determinations

All cell lines were cultured in RPMI 1640 medium with 5% fetal calf serum, to which were added 100 IU penicillin mL^−1^, 100 μg streptomycin mL^−1^, and 2 mM glutamine, at 37 °C in an air/carbon dioxide atmosphere (95:5, *v*/*v*). Cytotoxicity was determined by continuous drug exposure. Experiments were performed in 96-well microtiter plates (10^5^ cells/mL for NSCLC-N6-L16 and 2 × 10^−4^ cells mL^−1^ for A549). Cell growth was estimated by a colorimetric assay based on the conservation of tetrazolium dye (MTT) to a blue formazan product by live mitochondria. Eight repeats were performed for each concentration. Control growth was estimated from eight determinations. Vinorelbine ditartrate salt hydrate was used as positive control in experiments. Optical density at 570 nm corresponding to solubilized formazan was read for each well on a Titertek Multiskan MKII [29].

#### 3.7.2. Microorganism Cultures

A panel of microorganisms, including two Gram-positive bacteria, *Staphylococcus aureus* (ATCC 25923) and *Staphylococcus epidermidis* (ATCC 12228); four Gram-negative bacteria, *Escherichia coli* (ATCC 25922), *Enterobacter cloacae* (ATCC 13047), *Klebsiella pneumoniae* (ATCC 13883), and *Pseudomonas aeruginosa* (ATCC 227853); as well as three pathogenic fungi, *Candida albicans* (ATCC 10231), *Candida tropicalis* (ATCC 13801), and *Candida glabrata* (ATCC 28838) were used. The standard antibiotics netilmicin and amphotericin B were used in order to control the tested bacteria and fungi [29].

#### 3.7.3. Determination of Antimicrobial Activity

The antimicrobial activities of the crude extracts and the isolated compounds were determined using the agar dilution technique [27]. For all assays, stock solutions of the tested samples were prepared at 10 mg/mL. Serial dilutions of the stock solutions in broth medium (100 μL of Mueller Hinton broth or on Sabouraud broth for the fungi) were prepared in a microtiter plate (96 wells). Then, 1 μL of the microbial suspension (the inoculum, in sterile distilled water) was added to each well. For each strain, the growth conditions and the sterility of the medium were checked and the plates were incubated as stated above. MICs were determined as the lowest concentrations preventing visible growth. The standard antibiotics netilmicin, amoxicillin, and clavulanic acid (at concentrations 4–88 μg/mL) were used in order to control the sensitivity of the tested bacteria, while amphotericin B and 5-fluocytocine (at concentrations 0.4–1 μg/mL) were used as controls against the tested fungi (Sanofi Diagnostics Pasteur). For each experiment, any pure solvent used was also applied as blind control. The experiments were repeated three times and the results were expressed as average values.

## 4. Conclusions

*Rindera graeca*, a rare Greek endemic plant, could be efficiently micropropagated in in vitro cultures. The in vitro cultivated plant material was studied phytochemically for the first time and several secondary metabolites were identified, consisting of six caffeic acid derivatives, one flavonol glycoside (quercetin-3-rutinoside-7-rhamnoside), and a rare furano-naphthoquinone (rinderol).

The chemical profiles of in vitro cultures and the plant in nature were comparable, rich in phenolic metabolites, and the same PAs were identified. It is noteworthy that the naphthoquinone rinderol was detected and isolated only from in vitro cultures, while the quercetin diglycoside rutin was absent.

n-Hexane and methanol extracts, as well as pure rinderol, were tested against human non-small-cell bronchopulmonary carcinoma cell line NSCLC-N6-L16 and the epidermoid lung cancer cell line A549, and rinderol revealed strong activity, which was further evaluated with continuous growth kinetics for its potential mode of action. The results showed that rinderol did not act in a comparable manner against both tested cancer cell lines, as it exerted a dose-dependent cytostatic activity only against NSCLC-N6-L16. Due to the very interesting results, further antiproliferative studies are under development.

All extracts and rinderol studied for their antibacterial activity showed strong activity against Gram-positive *S. aureus* and *S. epidermidis* strains.

*Rindera graeca* in vitro cultures could serve as a reasonable source of very important caffeic derivatives, bioactive quercetin-triglycoside, and the unique naphthoquinone rinderol. The present study confirms the beneficial potential of in vitro root cultures for the production of essential secondary metabolites, and indicates that they could serve as a source towards a sustainable approach for the production of bioactive compounds for further research and applications.

## Figures and Tables

**Figure 1 plants-10-00834-f001:**
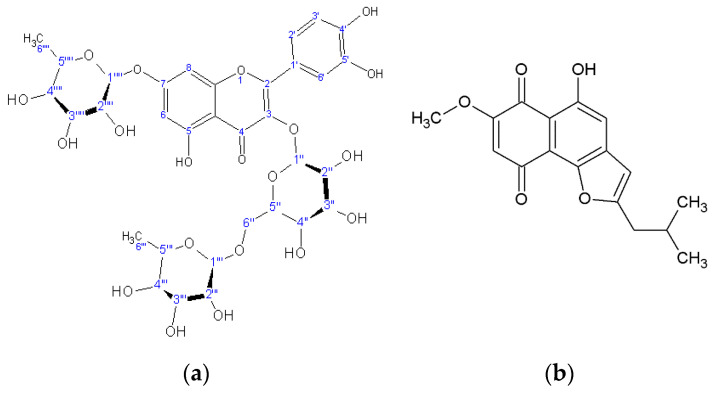
Chemical structure of quercetin 3-rutinoside-7-rhamnoside (**a**) and rinderol (**b**).

**Figure 2 plants-10-00834-f002:**
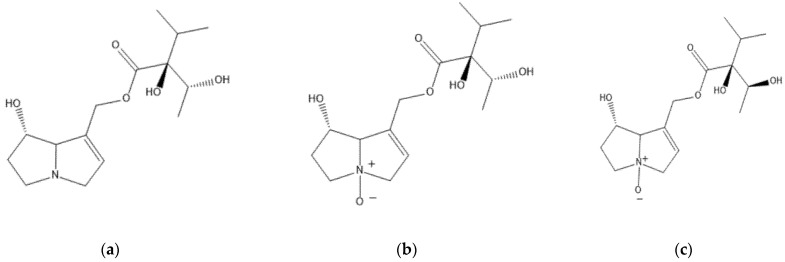
The structures of identified PA/PANOs: (**a**) echinatine, (**b**) echinatine N-oxide, (**c**) rinderine N-oxide.

**Figure 3 plants-10-00834-f003:**
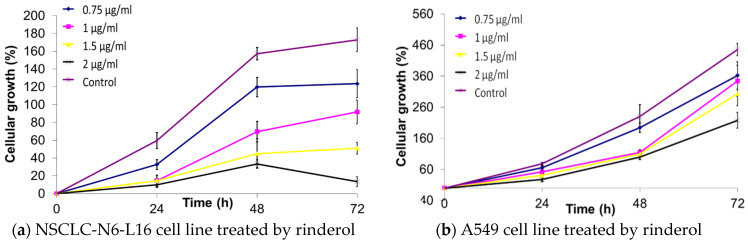
Result of continuous growth kinetics of NSCLC-N6-L16 (**a**) and the line A549 (**b**) cell lines after treatment with different concentrations of rinderol in a period of 72 h.

**Figure 4 plants-10-00834-f004:**
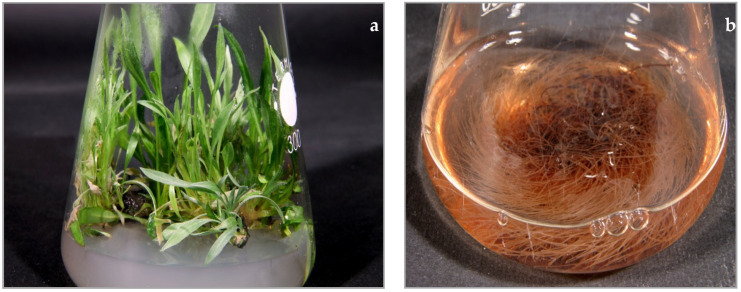
*Rindera graeca* in vitro culture: (**a**) shoots cultivated on solid DCR medium supplemented with BAP 0.5 mg/L; (**b**) hairy roots maintained in liquid DCR medium.

**Table 1 plants-10-00834-t001:** Comparison of the chemical analyses between in vitro cultures of hairy root in vitro cultures of shoots and aerial parts of the plant in nature.

Detected/Isolated Metabolites	*m*/*z*	In Vitro Cultures of Hairy Roots	In Vitro Cultures of Shoots	Natural Aerial Parts	Refs.
Lithospermic acid	537.1161	√	√	-	[1]
Lithospermic acid B	717.1497	-	-	√	[8]
Rosmarinic acid	359.0797	√	√	√	[8]
Chlorogenic acid	353.0852	-	√	√	[8]
Caffeic acid	179.0321	-	√	√	[8]
Salvianolic acid A	493.0985	-	-	√	[8]
Rabdosiin disodium salt	741.1369	-	√	√	[5,8]
Rabdosiin	717.1391	-	√	√	
Rinderol	301.1082	√	-	-	[6]
Quercetin 3-rutinoside	609.1445		-	√	[8]
Quercetin 3-rutinoside-7-rhamnoside	755.2013	√	√	√	
Echinatine	300.1812	√	√	√	[8,10,19]
Echinatine N-oxide	316.1768	√	√	√	
Rinderine N-oxide	316.1756	√	√	√	

**Table 2 plants-10-00834-t002:** Results of cytotoxicity on human lung cancer cell lines (IC50) of *R. graeca* in vitro cultivated shoots and hairy roots extracts, pure rinderol, and vinorelbine.

IC50 (µg/mL)	A549	NSCLC-N6-L16
n-Hexane extract of in vitro hairy roots	12.7 ± 0.4	13.6 ± 0.80
Methanol extract of in vitro shoots	16.6 ± 3.1	22.7 ± 0.20
Rinderol	0.9 ± 0.12	1.2 ± 0.10
Vinorelbine	0.04 ± 0.94	0.001 ± 0.04

**Table 3 plants-10-00834-t003:** Results of antimicrobial activities (minimum inhibitory concentration (MIC) values) of all tested extracts and pure metabolites (µg/mL).

	*S. aureus*	*S. epidermidis*	*P. aeruginosa*	*K. pneumoniae*	*E. cloacae*	*E. coli*	*C. albicans*	*C. tropicalis*	*C. glabrata*
Hexane extract of in vitro hairy roots	0.68	0.50	1.90	2.25	2.12	1.98	1.98	1.70	1.55
MeOH extract of in vitro shoots	1.18	1.12	2.50	2.97	2.45	2.42	3.10	2.65	2.38
Rinderol	1.2 × 10^−2^	0.98 × 10^−2^	12 × 10^−2^	10 × 10^−2^	0.95 × 10^−2^	0.8 × 10^−1^	1.10	0.97	0.80
MeOH extract aerial parts	0.88	0.76	2.3	2.50	2.20	2.33	2.87	2.48	2.27
Netilmicin	3.5 × 10^−3^	3.8 × 10^−3^	7.4 × 10^−3^	8.3 × 10^−3^	7.2 × 10^−3^	3.45 × 10^−3^	-	-	-
Amoxicillin	2 × 10^−3^	1.8 × 10^−3^	2 × 10^−3^	2 × 10^−3^	2.5 × 10^−3^	2 × 10^−3^	-	-	-
5-Flucytosine	-	-	-	-	-	-	0.15 × 10^−3^	0.95 × 10^−3^	9.5 × 10^−3^
Amphotericin B	-	-	-	-	-	-	1.20 × 10^−3^	0.49 × 10^−3^	0.5 × 10^−3^

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
