# Peer review of "Chemical Profile and Screening of Bioactive Metabolites of Rindera graeca (A. DC.) Bois. & Heldr. (Boraginaceae) In Vitro Cultures"

_plants, 2021, doi:10.3390/plants10050834_

Round 1

Reviewer 1 Report

The manuscript from Konstantia Graikou et al, describes a metabolomics analysis of Rindera  graeca and  the potential biological effect of specific molecules present in the plant extract.

The article is suitable for Plants. However, there are minor flaws surrounding this piece of work.

  • I would recommend the authors to have a spelling check and to use proper chemistry terminology. For example the sentence” positive charge ESI ion source was used” please use the correct terminology” the ms scan were acquired in positive ionization mode”.
  • It is unclear from the text if the compounds were eluted isocratically or using a gradient.
  • Please provide m/z values for all the identified compounds.
  • Please provide a chromatogram as supplementary figure for compound separation.
  • Please provide NMR spectra for the major compounds and the chemical shifts in ppm in the supplemental material.

Author Response

The manuscript from Konstantia Graikou et al, describes a metabolomics analysis of Rindera  graeca and  the potential biological effect of specific molecules present in the plant extract.

The article is suitable for Plants. However, there are minor flaws surrounding this piece of work.

  • I would recommend the authors to have a spelling check and to use proper chemistry terminology. For example the sentence” positive charge ESI ion source was used” please use the correct terminology” the ms scan were acquired in positive ionization mode”.

The manuscript has been checked and several sentences have been changed throughout the text

  • It is unclear from the text if the compounds were eluted isocratically or using a gradient.

For the methanol extract of in vitro shoots an isocratic 9:1 MeOH/CH2Cl2 mobile phase was used while for the n-hexane extract of in vitro hairy roots cyclohexane:CH2Cl2 and CH2Cl2:MeOH step gradient was used.

  • Please provide m/z values for all the identified compounds.

The m/z values for all the identified compounds were added in the Table 1

  • Please provide a chromatogram as supplementary figure for compound separation.
  • Please provide NMR spectra for the major compounds and the chemical shifts in ppm in the supplemental material.

A supplementary file with chromatograms, TLC plate, LC/MS data, NMR spectra and tables with chemical shifts has been created and added to the submission.

Reviewer 2 Report

The MS entitled "Chemical Profile and Screening of Bioactive Metabolites of Rindera graeca (A. DC.) Bois. & Heldr. (Boraginaceae) in vitro Cultures" present novel and interesting study investigating bioactive metabolites of Rindera graeca (rare endemic plant) in vitro culture. Very exciting findings are reported concerning the activity of rinderol (found only in in vitro hairy roots) against some Gram positive bacteria and antiproliferative effect against one type of human non-small-cell bronchopulmonary carcinoma cell line (NSCLC-N6-L16).  

However, the manuscript need some improvements before publication.

The authors need to include the detection results for the eight identified metabolites like in Ref [8] as in the current MS they were identified in in vitro culture. Probably the results were the same as in previous papers but this MS should show the results as the reader need to see them together with the other reported results. In addition, Method section have part "analysis of PA", "Liquid Chromatography/Mass Spectrometry (LC/MS) analysis of PAs", and "Nuclear Magnetic Resonance (NMR) analysis" but the data in "Results and discussion" section are missing.

Please, check Table 1 for mistakes, as by my oppinion nicotiflorin and kaempferol 3-robino-side-7-rhamnoside were not mentioned in Ref [8].

It is important some introduction sentences to be altered as they were used almost the same way in a previous paper [8].

The sentence “In addition, is a type of lung cancer difficult to detect and quickly forms metastasis which makes surgery almost impossible.” (line 122-123) need to be improved.

Please, include in the caption of Table 2 “ the human lung cancer cell lines” as the tables and figures  need to be self-explanatory. The heading of Table 2 need to include "pure rinderol and vinorelbin".

Figure 3: The heading should be changed in order to be self-explanatory and axis (Y) should be improved.

"Ruinderol"(line 158) must be changed to rinderol.

Figure 4: check if a) and b) are exchanged by mistake. Improve the formatting of Fig. 4b (the label ”b” is not well visible and the photo should be in the same line with the “a”-panel).

In my opinion the MS will benefit if the conclusions are more compact and concise.

Author Response

The MS entitled "Chemical Profile and Screening of Bioactive Metabolites of Rindera graeca (A. DC.) Bois. & Heldr. (Boraginaceae) in vitro Cultures" present novel and interesting study investigating bioactive metabolites of Rindera graeca (rare endemic plant) in vitro culture. Very exciting findings are reported concerning the activity of rinderol (found only in in vitro hairy roots) against some Gram positive bacteria and antiproliferative effect against one type of human non-small-cell bronchopulmonary carcinoma cell line (NSCLC-N6-L16).  

However, the manuscript need some improvements before publication.

The authors need to include the detection results for the eight identified metabolites like in Ref [8] as in the current MS they were identified in in vitro culture. Probably the results were the same as in previous papers but this MS should show the results as the reader need to see them together with the other reported results.

A supplementary file with chromatograms, TLC plate, LC/MS data, NMR spectra and tables with chemical shifts has been created and added to the submission.

In addition, Method section have part "analysis of PA", "Liquid Chromatography/Mass Spectrometry (LC/MS) analysis of PAs", and "Nuclear Magnetic Resonance (NMR) analysis" but the data in "Results and discussion" section are missing.

In Results several sentences have been added according to the suggestion.

Please, check Table 1 for mistakes, as by my oppinion nicotiflorin and kaempferol 3-robino-side-7-rhamnoside were not mentioned in Ref [8].

Table 1 has been corrected as by mistake there were some additional compounds, which are not related to this study.

It is important some introduction sentences to be altered as they were used almost the same way in a previous paper [8].

The Introduction has been modified according to the suggestion

The sentence “In addition, is a type of lung cancer difficult to detect and quickly forms metastasis which makes surgery almost impossible.” (line 122-123) need to be improved.

The sentence has been improved as “. In addition, is a type that is difficult to detect and readily gives metastasis which makes surgery almost impossible.”

 Please, include in the caption of Table 2 “ the human lung cancer cell lines” as the tables and figures  need to be self-explanatory. The heading of Table 2 need to include "pure rinderol and vinorelbin".

The Table 2 legend has been changed to “Results of cytotoxicity on human lung cancer cell lines (IC50) of R. graeca in vitro cultivated shoots and hairy roots extracts, pure rinderol and vinorelbine”

Figure 3: The heading should be changed in order to be self-explanatory and axis (Y) should be improved.

The Figure 3 heading has been improved to: “Result of continuous growth kinetics of NSCLC-N6-L16 (a) and the line A549 (b) cell lines after treatment with different concentrations of rinderol in a period of 72hours” 

"Ruinderol"(line 158) must be changed to rinderol.

It has been corrected

Figure 4: check if a) and b) are exchanged by mistake. Improve the formatting of Fig. 4b (the label ”b” is not well visible and the photo should be in the same line with the “a”-panel).

The figure 4 has been improved and the legend has been corrected as the a) and b) were exchanged by mistake. The correct one is :”Rindera graeca in vitro culture: a) shoots cultivated on solid DCR medium supplemented with BAP 0.5 mg/L; b) hairy roots maintained in liquid DCR medium”

 In my opinion the manuscript will benefit if the conclusions are more compact and concise.

The Conclusion part has been modified in order to be more concise.

Reviewer 3 Report

The study presented in the manuscript include several chromatographic analysis of plant extract, however there is a lack of results of such determinations. Could the authors consider presenting more detailed data (like images of TLC plate, LC/MS data, NMR spectra) for example in supplementary materials? 

The title of the Table 2 should be reworded since it shows results of cytotoxicity determination presented as IC50 not results of IC50.

Author Response

The study presented in the manuscript include several chromatographic analysis of plant extract, however there is a lack of results of such determinations. Could the authors consider presenting more detailed data (like images of TLC plate, LC/MS data, NMR spectra) for example in supplementary materials? 

A supplementary file with chromatograms, TLC plate, LC/MS data, NMR spectra and tables with chemical shifts has been created and added to the submission.

The title of the Table 2 should be reworded since it shows results of cytotoxicity determination presented as IC50 not results of IC50.

The Table 2 legend has been changed to “Results of cytotoxicity on human lung cancer cell lines (IC50) of R. graeca in vitro cultivated shoots and hairy roots extracts, pure rinderol and vinorelbine”

Round 2

Reviewer 3 Report

I accept the introduced corrections, however there are no references to the data from supplementary material. It should be completed before publication.

Author Response

References to the data from supplementary material have been added. The data are from our team works and they are already published.

References

  1. Tufa, T., Damianakos, H., Zengin, G., Graikou, K., Chinou, I. Antioxidant and enzyme inhibitory activities of disodium rabdosiin isolated from Alkanna sfikasiana Tan, Vold and Strid. S Afr J Bot, 2019, 120, 157-162.
  2. Ganos, C.; Aligiannis, N.; Chinou, ; Naziris, N.; Chountoulesi, M.; Mroczek, T.; Graikou, K. Rindera graeca (Boraginaceae) phytochemical profile and biological activities. Molecules, 2020, 25, 3625 doi:10.3390/molecules25163625
  3. Jeziorek, M., Damianakos, H., Kawiak, A., Laudy A.E., Zakrzewska, K., Sykłowska-Baranek, K., Chinou, I., Pietrosiuk, A. Bioactive rinderol and cynoglosol isolated from Cynoglossum columnae in vitro root culture. Ind. Crops Prod. 2019, 137, 446-452. doi: 10.1016/j.indcrop.2019.04.046
  4. Damianakos, H.; Jeziorek, M.; SykÅ‚owska-Baranek, K.; Buchwald, W.; Pietrosiuk, A.; Chinou, I. Pyrrolizidine alkaloids from Cynoglossum columnae (Boraginaceae). Phytochemistry Letters 2016, 15, 234–237